# Analytical Aspects of Distributed MPC Based on an Occupancy Grid for Mobile Robots

**Tobias Sprodowski** [1,*] and **Jürgen Pannek** [1,2]

[1]   Faculty of Production Engineering, Hochschulring 20, University of Bremen, 28359 Bremen, Germany
[2]   BIBA Bremer Institut für Produktion und Logistik GmbH, Hochschulring 20, 28359 Bremen, Germany; pan@biba.uni-bremen.de
*   Correspondence: spr@biba.uni-bremen.de; Tel.: +49-421-218-50097

**Abstract:** In this paper, we evaluate theoretical aspects of a distributed system of noncooperative robots controlled by a distributed model predictive control scheme, which operates in a shared space. Here, for collision avoidance, the future predicted state trajectories are projected on a grid and exchanged via discrete cell indexes to reduce the communication burden. The predicted trajectories are obtained locally by each robot and carried out in the continuous space. Therefore, the quantisation does not impose the quality of the solution. We derive sufficient conditions to show convergence and practical stability for the distributed control system by using an idea of a temporary roundabout derived from crossing patterns of street traffic rules, which is established in a fixed and flexible circle size. Furthermore, a condition for the sufficient prediction horizon length to recognise necessary detours is presented, which is adapted for the occupancy grid. The theoretical results match with the trajectory patterns from former numerical simulations, showing that this pattern is naturally chosen as an overall solution.

**Keywords:** distributed model predictive control; quantisation; stability; prediction horizon length; mobile robots

## 1. Introduction

Today, control problems with high dynamics have been of high interest in production and logistic processes or in traffic scenarios. In manufacturing processes, the technical advances have increased the utilisation of robots, especially in assembly lines in the last decades [1]. In logistic processes, autonomous, connected pallet carriers for material have increasingly been used to supply production lines with material. In common traffic scenarios, autonomous connected cars (or mobile robots) have aggregated different research fields, e.g., sensor fusion, modelling, and handling communication. One central issue is to be able to calculate a feasible solution which ensures safety in terms of collision avoidances. Examples for robotic systems can be found in formation control of robots [2], exploration of unknown environment to achieve an optimal covering with limited number of robots [3], or real-time control of distributed helicopters [4]. All these distributed systems are linked by common properties and arising problems: complexity and availability of information. Concerning complexity, these system have to be handled in a distributed manner. One approach is to use multi-agent systems to decrease the complexity and to enable the distributed system to calculate a solution sufficiently fast. The available computational capabilities on distributed systems like robots or cars ease this approach.

Additionally, the distributed system is relying on the requirement of an information exchange: In contrast to a central system, where the information about the full system state is globally available, here, each agent has to rely on local information (decentralised system) or some communication methods have to be implemented such that the agents are able to gather information from neighbours

or aggregated information about the full system state. If stability is taken into account, such information exchange is essential to achieve and guarantee that stability could be kept; see References [5–7].

The scenario here describes a group of robots in a shared operation space (spacial set), which shall be steered to individual targets while ensuring collision avoidance. This is implemented in a noncooperative control scheme, where each robot optimises a local optimal control problem (OCP) to minimise a cost function regarding the current distance to an individual target. Therefore, additional communication among the robots is necessary to inform the other robots by the calculated prediction to let them regard the collision avoidance via coupling constraints in the OCP. On the other hand, cooperative control schemes are targeting a common goal, which could be imposed for example via a common objective; see Reference [8] for flight formation of robots. The optimisation is executed in the spacial set, while for the communication among the robots, a quantisation of the continuous space to a grid with equidistant cells is established. The predicted trajectories are quantised obtaining grid indices and used to communicate them between the robots. This approach, introduced in References [9,10], allows to keep the problem suitable for optimisation algorithms in a continuous setting while taking advantage in the communication reduction via discretisation. The optimisation and therefore the communication is carried out in a sequential manner according to Reference [11]; the future predicted state trajectories of each robot (here, discrete occupancy tuples) are communicated iteratively for each time instant. While in the former work [9,10] conditions considering initial and recursive feasibility were examined, the focus here is set to derive conditions for convergence and practical stability of the overall system.

We focus on theoretical properties of this approach to ensure two aspects: deadlock-free execution and convergence. Considering convergence properties in Distributed Model Predictive Control (DMPC), many advances were made in the last years, c.f. Reference [12]. For cooperative schemes, in Reference [13], the authors introduced DMPC with bounded disturbances where the bound of the disturbance is time varying. For both decentralised and distributed versions, feasibility was shown. Several cooperative control schemes were utilised to show these properties: the authors of Reference [5] used terminal costs to speed up the convergence rate while assuming bounded interactions among the subsystems. In Reference [14], a cooperative control was applied where the subsystems optimise sequentially with predetermined terminal regions and cost functions to achieve feasibility and convergence. The authors of Reference [15] added a global stability constraint to show convergence, which is locally impaired by each subsystem. In Reference [16], the authors address linear networked systems in cooperative control regarding the consensus via static couplings, which are imposed on the input of each subsystem from the output of other systems. For gradient-based DMPC schemes based on dual decomposition, an upper bound on the system dynamics is used to improve the convergence rate [17]. The results were extended in Reference [18] to general DMPC problems, and convergence is shown via a stopping criterion which computes lower and upper bounds when a sufficient control is found to keep the system in the desired state. Considering noncooperative schemes, a global objective was completely decomposed in Reference [19] and applied on network traffic control,. Moreover, the control values of all affected neighbours were considered in a proposed communication protocol to obtain a fixed decision order for execution. In Reference [20], an iterative exchange of information in each time instant was not needed, as reference trajectories are provided for each subsystem while convergence is ensured via terminal constraints. The authors of Reference [21] utilised the scheme of Reference [11] to show feasibility and convergence without using terminal costs or constraints. Therefore, asymptotic controllability of the distributed system was assumed to derive stability properties. Instead of using a fixed optimisation order, in Reference [22], a dynamic deordering rule was presented to iteratively change the order dynamically on an abstract criterion. Convergence was shown based on a relaxed Lyapunov condition. A leader–follower system with flexible leader was considered in Reference [23] with an arbitrary chosen convex hull. The sequential order of the execution of optimisation is fixed, while small perturbations were incorporated. Regarding slow dynamics, in Reference [24] a quantisation scheme was applied for the communication among subsystems which

allow slow variation of the quantisation interval. Here, the quantisation parameters are exchanged among the subsystems to establish terminal regions which ensured that, from a warm start initial condition, the equilibrium was reached within finite time.

Other approaches to stabilise nonlinear systems used Lyapunov-based controllers to keep the system stable with tight state and input constraints, while relaxations on the states were possible via a switched control [25]. On the distributed level, in Reference [26], the Lyapunov controllers were executed in either a sequential or iterative manner while stability was ensured by a cost decrease constraint. In Reference [27], the authors relaxed the Lyapunov scheme via a time delay to allow for a temporary increase of costs. The authors in Reference [28] applied a Lyapunov controller with a global bounded state feedback controller for a tracking problem of a non-holonomic robot.

In these articles, the convergence or stability aspects were shown by utilising either a Lyapunov function or a controller, which guarantees the decrease of costs in any time instant, or of terminal constraints or costs, which then forces the predicted trajectory to achieve an end point or region, or to steer that by a terminal cost penalty or a connective constraint, ensuring that, e.g., a minimum distance or explicit bounds on the optimal value function were used to calculate the suboptimality degree of a finite horizon controller. As the considered scenario here is noncooperative and as a Lyapunov function for one robot may be infeasible if one robot has to increase its costs by taking a detour and if terminal regions or costs will restrict the freedom of feasible trajectories, we use a weaker assumption, which is similar to the movement pattern from former numerical results based on the pattern of a roundabout; see Figures 4 and 7 in Reference [10].

We focus on a noncooperative setting as each robot exhibits an individual target without imposing terminal costs or constraints following Reference [21]. In this article, we derive sufficient conditions to avoid blockings or deadlocks with the assumption that the robots optimise in a fixed order in the spacial set with quantised communication. We avoid imposing Lyapunov conditions for the cost function, which would either impose additional constraints for the OCP or terminal costs. Hence, we implement a switched control, which is inspired from street traffic rules for a roundabout structure. If the robots are below a certain distance to each other, a circle with the mean-based middle point based on the end points of the predicted trajectories of the affected systems is created. Each participating system calculates individual intersection points with this circle and follows the circular curve via a circular control law until the original control law could be reinstated again. Moreover, as a second aspect, we derive a sufficient condition for the prediction horizon length based on a given cost function to examine the sufficient length, which allows the robot to recognise the decrease of costs to take the detour instead of holding their positions probably infinitely long.

The paper is structured as follows: In the subsequent section, the problem is stated with the model of the robots including state and control constraints. We recap the construction of the occupancy grid shortly and give an outline on the construction of the constraints including a safety margin regarding the quantised communication. Then, these parts are assembled to define an OCP in Section 3, which is incorporated in the DMPC scheme on each robot. The theoretical contributions concerning a sufficient prediction horizon length and the convergence to achieve practical stability are presented in Section 4. For the convergence part, we consider two possibilities: First a fixed circle size is considered to possibly include all robots on the circle. Second, we extend this approach to a flexible circle, which considers the maximum size without interfering with targets of other robots. To complete the article, a short conclusion and outlook on future extensions is given.

**Notation:** $\mathbb{R}$ and $\mathbb{N}$ denote the real and natural numbers, respectively. $\mathbb{N}_0 := \mathbb{N} \cup \{0\}$ represents the nonnegative integers, and $\mathbb{R}_{\geq 0}$ represents the nonnegative real numbers. For integers $a, b \in \mathbb{N}_0$ with $a \leq b$, the expression $[a : b]$ denotes the sequence $\{a, a+1, \cdots, b\}$. Moreover, for a vector $x \in \mathbb{R}^n$, $n \in \mathbb{N}$, we define the infinity norm $\|x\|_\infty := \max_{i \in [1:n]} |x_i|$.

## 2. Problem

Based on Reference [10], we introduce the problem and necessary definitions for the robots, occupancy grid, and constraint construction shortly: Here, we utilise a distributed system of $P \in \mathbb{N}$ mobile robots. The robots follow a time discrete model stated as a control-affine system:

$$z_p^+ = f(z_p, u_p) := f_0\left(z_p\right) + \sum_{i=1}^{m} f_i\left(z_p\right) u_{p,i}, \qquad p \in [1:P] \tag{1}$$

with smooth vector fields $f_0$ and $f_i \colon \mathbb{R}^d \to \mathbb{R}^d$, $d \geq 2$, $i \in \{1, \ldots, m\}$, $d, m \in \mathbb{N}$ where, for each vector field $f_i$, $i \in \{1, \ldots, m\}$, one control value $u_{p,i}$ is imposed. A successive state of a robot is denoted by $z_p^+$ and determined by $f_i$ based on its current state $z_p$ and imposed control $u_p$. The state is represented here without loss of generality by the planar coordinates $(x_p, y_p)^\top = z_p \in \mathbb{R}^d$. Constraints for state and control are given by

$$z_p \in Z := [-\bar{x}, \bar{x}] \times [-\bar{y}, \bar{y}] \times \widetilde{Z} \qquad \text{with } \widetilde{Z} \subseteq \mathbb{R}^{d-2} \text{ and } \bar{x}, \bar{y} > 0, \tag{2}$$

$$u_p \in U \subset \mathbb{R}^m, \tag{3}$$

respectively, with $U$ containing the origin.

Assuming that our systems in Equation (1) are driftless ($f_0 \equiv 0$), describing the kinematic model of a mobile robot, we assume that robots may immediately stop by the following assumption:

**Assumption 1** (Immediate Hold). *Each robot can come to an immediate hold:*

$$\forall z_p \in Z \quad \exists \bar{u}_p \in U \quad : \quad z_p = f(z_p, \bar{u}_p). \tag{4}$$

Note that delays of actuator dynamics, which might invalidate this assumption are not taken into account. For a finite horizon $N$ and individual initial positions $z_p(0)$, for each robot $p$, a finite control sequence is defined by

$$\mathbf{u}_p = (u_p(0), u_p(1), \ldots, u_p(N-1)),$$

which is utilised to formulate the predicted state trajectory of robot $p$ as

$$\left(z_p^u\left(0; z_p^0\right), z_p^u\left(1; z_p^0\right), \ldots, z_p^u\left(N; z_p^0\right)\right).$$

### 2.1. Occupancy Grid and Quantised Communication

Based on a spacial set given by $[-\bar{x}, \bar{x}] \times [-\bar{y}, \bar{y}]$, we construct the occupancy grid by partitioning the spacial set into equidistant cells: $\mathcal{G} := [0 : a_{\max} - 1] \times [0 : b_{\max} - 1] \subset \mathbb{N}_0^2$ with cell width (= height) $c$. The partitioning is defined by $\bar{x}$ and $\bar{y}$, both multipliers of $c$ with

$$a_{\max} = \frac{2\bar{x}}{c} \qquad \text{and} \qquad b_{\max} = \frac{2\bar{y}}{c}$$

where $\bar{x}$ and $\bar{y}$ are both multiples of $c$. Furthermore, we assign an index $(a, b) \in \mathcal{G}$ to each cell; see Figure 1 for details.

For the quantisation of communication among the robots, we define $q \colon Z \to \mathcal{G}$:

$$q\left(z_p\right) = (a_p, b_p) := \left(\left\lfloor \frac{x_p + \bar{x}}{c} \right\rfloor, \left\lfloor \frac{y_p + \bar{y}}{c} \right\rfloor\right), \tag{5}$$

to map a state $z_p$ onto the grid $\mathcal{G}$. Now, this enables robot $p$ to broadcast the occupied cells instead of a continuous predicted state trajectory as a sequence of *occupancy tuples* $\mathcal{I}_p(n) \in (\mathbb{N}_0 \times \mathcal{G})^{N+1}$ at time $n$ with

$$\mathcal{I}_p(n) := \left( q\left( z_p^u\left( n+k; z_p^0 \right) \right) \right)_{k=0}^{N}.$$

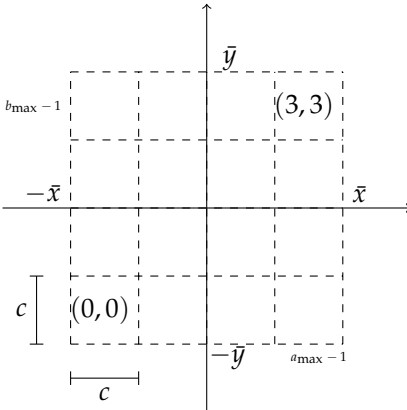

**Figure 1.** Example of a grid $\mathcal{G}$ with $a_{\max} = b_{\max} = 4$ from Reference [9].

All received occupancy grid tuples $\mathcal{I}_p(n)$ from the other robots $q$ with $q \in [1:P]\setminus\{p\}$ are assembled to

$$i_p(n) := \left( \mathcal{I}_1(n), \ldots, \mathcal{I}_{p-1}(n), \mathcal{I}_{p+1}(n), \ldots, \mathcal{I}_P(n) \right) \tag{6}$$

and denoted as the *occupancy grid*. To convert a quantised state back to obtain the spacial coordinates, we require the backward mapping $h\colon \mathcal{G} \to \mathbb{R}^2$ by

$$h(a,b) = \left( (a+0.5)\,c - \bar{x}, (b+0.5)\,c - \bar{y} \right)^{\top} \tag{7}$$

which maps a cell index $(a,b) \in \mathcal{G}$ to the spatial coordinates of the cell centre ignoring in both cases the time index $n$. This is then used in the following subsection to construct the coupling constraints, which are later used in the OCP of each robot.

## 2.2. Construction of the Constraints

The mobile robots share a collective operation space defined by $Z$. As their trajectories may cross each other, we have to ensure collision avoidance regarding two aspects: The first aspect is a minimum cell size of the occupancy grid to prevent robots from skipping cells or swapping cells with another robot. As the dynamics of the robots is discretised with a fixed sampling step, an overlapping of two trajectories may occur within two time instants, leading to an unnoticed crossing of robots. Hence, a minimum cell size to prevent this is imperative. The second aspect stems from a necessary physical distance among the robots and the adherence of the constraints in the discrete time instants and in between two successive time instants.

**Cell size**: In Reference [10], we proposed a minimum cell size $\underline{c}$ based on the maximum control value which can be imposed on the robot. This minimum cell size has to cover all possible moves of the robot in $Z$; therefore, it is lower bounded for each vector field $f_0$ and $f_i$ with $i \in \{1, \ldots, m\}$

$$\underline{c} := \max_{u_p \in U} \left\{ c_{f_0} + \sum_{i=1}^{m} c_{f_i} |u_{p,i}| \right\} \tag{8}$$

where each $c_i$ with $i \in \{0, \dots, m\}$ covers the dimensions with

$$c_{f_0} := \max_{z_p \in Z} \left\{ \max_{j \in \{1,2\}} | (f_0(z_p) - z_p)_j | \right\} \text{ and } c_{f_i} := \max_{z_p \in Z} \left\{ \max_{j \in \{1,2\}} |f_{i,j}(z_p)| \right\}.$$

Here, obeying the kinematic model of the robot, this is sufficient to avoid skipping cells.

**Safety margin in discrete time instants**: Considering the safety margin, we have to ensure that each robot keeps a physical minimum distance to other robots, which is given by $d_{min}$. As the minimum cell size can be smaller or larger then the necessary physical distance among the robots, we define the safety margin as follows:

**Definition 1** (Safety Margin). *The safety margin among any two robots $p, q \in P, p \neq q$ is given by the necessary physical minimum distance $d_{min}$ and the given minimum cell size $\underline{c}$ such that*

$$\left\| \begin{pmatrix} x_p \\ y_p \end{pmatrix} - \begin{pmatrix} x_q \\ y_q \end{pmatrix} \right\|_\infty \geq \max\{d_{min}, \underline{c}\} \tag{9}$$

*holds. Here, $(x_i, y_i)^\top = z_i, i \in \{p, q\}$ denote the planar positions of the robots.*

Furthermore, we have to ensure that this safety margin holds over all predicted states $z_p^u \left( k; z_p^0 \right)$ with $k \in [1 : N]$ of one robot $p$ to any other robot $q$. As the information of the states of robot $q$ is given via the quantised occupancy tuples $\mathcal{I}_q (n) (k) := q \left( z_q^u(n + k; z_q^0) \right)$ for $k \in [1 : N]$; for a given cell size $c \geq \underline{c}$, a robot may be in the centre of the communicated cell or on the boundary. Therefore, we obtain the maximal deviation of the quantised state to the continuous state of the robot by

$$\left\| \begin{pmatrix} x_q \\ y_q \end{pmatrix} - h \left( q \left( z_q \right) \right) \right\|_\infty \leq \frac{c}{2}.$$

This leads to the definition of the safety margin $\Psi$ regarding the quantised error:

**Proposition 1** (Quantised Prediction Safety Margin). *For a given cell size $c \geq \underline{c}$, minimum distance among robots $d_{min}$, and discrete time instants $k$ with $k \in [1 : N]$, the safety margin between two robots $p, q \in P, p \neq q$ based on the received quantised state $\mathcal{I}_q (n) (k)$ from robot $q$ is*

$$\left\| \begin{pmatrix} x_p (k) \\ y_p (k) \end{pmatrix} - h \left( \mathcal{I}_q (n) (k) \right) \right\|_\infty \geq \max \{d_{min}, \underline{c}\} + \frac{c}{2} =: \Psi. \tag{10}$$

*If this holds, then Definition 1 is satisfied with*

$$\begin{pmatrix} x_p \\ y_p \end{pmatrix} = \begin{pmatrix} x_p (k) \\ y_p (k) \end{pmatrix}, \quad \forall k \in [1 : N]. \tag{11}$$

The details of the proof utilising the triangle inequality are given in Reference [10].

**Proposition 2** (Intermediate Safety Margin). *Suppose that Equation (9) holds for two robots $p, q \in P, p \neq q$ for $c \geq \underline{c}$. Then, the minimum safety margin to prevent intermediate constraint violation in two successive time instants $k - 1, k \in [1 : N]$ is*

$$\Psi_q \left( \mathcal{I}_q (n) (k), \mathcal{I}_q (n) (k - 1) \right) = \begin{cases} \overline{\Psi} := \Psi + \underline{c}, & \text{if } \mathcal{I}_q (n) (k) \neq \mathcal{I}_q (n) (k - 1) \\ \underline{\Psi} := \Psi + \frac{c}{2} \cos \left( \frac{\pi}{4} \right), & \text{else.} \end{cases} \tag{12}$$

Again, the details of the proof are given in Reference [10].

**Derivation of the Constraints:** Now, we utilise the received occupancy tuples $\mathcal{I}_q(n), q \in [1 : P] \setminus \{p\}$ of the other robots to construct the coupling constraints including the safety margin $\Psi_q$. Here, the infinity norm is used as this approximate the geometry of the squared cells.

Based on the received information, we map the state back to the continuous spacial set via $h\left(\mathcal{I}_q(n)(k)\right)$. Given the safety margin $\Psi_q\left(\mathcal{I}_q(n)(k), \mathcal{I}_q(n)(k-1)\right)$, the coupling constraint for robot $p$ regarding robot $q$ is defined as

$$g_{q,k}^p := g\left(z_p^u\left(k; z_p^0\right), \mathcal{I}_q(n)(k), \mathcal{I}_q(n)(k-1)\right) \tag{13}$$

$$= \left\| \begin{pmatrix} x_p(k) \\ y_p(k) \end{pmatrix} - h\left(\mathcal{I}_q(n)(k)\right) \right\|_\infty - \Psi_q\left(\mathcal{I}_q(n)(k), \mathcal{I}_q(n)(k-1)\right) \geq 0 \tag{14}$$

for all $k \in [1 : N]$. Note that, although the infinity-norm describes the property of the occupancy grid very well, it is not differentiable. Hence, a gradient-free optimisation method should be used, which is in general more time consuming than gradient-based methods; see, e.g., Reference [29].

For all robots $q$ with $q \neq p$, the received information is condensed in $i_p(n)$. Based on the coupling constraints obeyed by robot $p$, the combined constraints are given by

$$G\left(z_p^u\left(k; z_p^0\right), i_p(n)(k), i_p(n)(k-1)\right) = \left(g_{1,k}^p, \ldots, g_{p-1,k}^p, g_{p+1,k}^p, \ldots, g_{P,k}^p\right) \tag{15}$$

for $g_{i,k}^p \geq 0$ with $i \in (1, \ldots, p-1, p+1, \ldots, P)$ and for $k \in [1 : N]$. If these constraints are satisfied, the quantised prediction safety margin based on the information $i_p(n)$ and in turn Definition 1 is ensured.

## 3. Distributed Model Predictive Control

To complete the requirements to state an OCP, we introduce a strict convex stage cost function $\ell_p : Z \times U \to \mathbb{R}_{\geq 0}$, which is a positive definite with respect to an individual target $z_p^* \in Z$ for each robot. In this distributed setting, each robot is steered by a controller which solves an OCP over a finite horizon length $N$ based on the measured state. Then, after solving its OCP, the first control value of the obtained control sequence is applied, and the procedure is repeated until a termination condition is met, i.e., the target $z_p^*$ is reached. To this end, the overall OCP is stated as

$$\min_{\mathbf{u}_p} J_p^N\left(\mathbf{u}_p; z_p^0, i_p(n)\right) := \sum_{k=0}^{N-1} \ell_p\left(z_p^u\left(k; z_p^0\right), u_p(k)\right) \tag{16}$$

subject to

$$
\begin{aligned}
z_p^u(k+1; z_p^0) &= f(z_p^u(k; z_p^0), u_p(k)), & k &\in [0 : N-1], \\
u_p(k) &\in U, & k &\in [0 : N-1], \\
G\left(z_p^u\left(k; z_p^0\right), i_p(n)(k), i_p(n)(k-1)\right) &\geq 0, & k &\in [1 : N], \\
z_p^u\left(k; z_p^0\right) &\in Z, & k &\in [1 : N], \\
z_p^* &\in Z, \\
z_p(0) &\in Z
\end{aligned} \tag{17}
$$

regarding the given constraints concerning the kinematic model, control, state constraints, and coupling constraints induced by the information of the other robots and by individual initial conditions and targets. By solving the OCP, an optimal control sequence

$$\mathbf{u}_p^* = \left( u_p^*(0), \dots, u_p^*(N-1) \right), \tag{18}$$

is obtained, where uniqueness is ensured by the strict convexity of the cost function $\ell_p$. We introduce the corresponding value function as $V_p^N \colon Z \times (\mathbb{N}_0 \times \mathcal{G})^{(P-1)(N-1)} \to \mathbb{R}_{\geq 0}^+$ via

$$V_p^N \left( z_p^0, i_p(n) \right) = J_p^N(\mathbf{u}_p^*; z_p^0, i_p(n)). \tag{19}$$

The DMPC scheme, which is executed by each robot based on the scheme by References [30,31], is presented in algorithmic form and is divided in an initialisation phase (Algorithm 1) and an execution phase (Algorithm 2).

---

**Algorithm 1** DMPC initialisation for the overall system

---

1: **Given** admissible states $z_p$ for initial states $z_p^0$ for all $p \in [1 : P]$
2: **for** $p = 1$ to $P$ **do**
3: 　　**for** $k = 1$ to $N$ **do**
4: 　　　　**Set** $\mathcal{I}_p(0)(k) := \left( k, q \left( z_p^0 \right) \right)$
5: 　　**end for**
6: 　　**Broadcast** $\mathcal{I}_p(0)$
7: 　　**Set** $w_p = false$
8: **end for**
9: **Set Q** $= \varnothing$

---

In Algorithm 1, every robot keeps its initial state in the first time instant and broadcasts this before the OCP is solved. **Q** and $w_p$ are needed later for the derivation of conditions for convergence and practical stability.

---

**Algorithm 2** DMPC execution for the overall system

---

1: **Call** Algorithm 1
2: **for** $n = 0, 1, \dots$ **do**
3: 　　**for** $p = 1$ to $P$ **do**
4: 　　　　**if** $n > 0$ **then**
5: 　　　　　　**Measure** $z_p(n)$
6: 　　　　　　**Receive** $\mathcal{I}_q(n)$ for $q \in [1 : P] \setminus \{p\}$
7: 　　　　　　**Assemble** $i_p(n)$ using Algorithm 3
8: 　　　　**else**
9: 　　　　　　**Receive** $\mathcal{I}_q(n)$ for $q \in [1 : P] \setminus \{p\}$
10: 　　　　　　**Set** $i_p(n)$ according to Equation (6)
11: 　　　　**end if**
12: 　　　　**Solve** OCP in Equation (16) and **Apply** $u_p^*(0)$
13: 　　　　**Broadcast** $\mathcal{I}_p(n)$
14: 　　**end for**
15: **end for**

---

As stated in Reference [10], the asymmetry of information has to be handled: Robot $p$ has information $\mathcal{I}_q(n)$ of the robots $q, q \in [1 : p-1]$, while for the successive robots $q, q \in [p+1 : P]$, the information of the last time instant $\mathcal{I}_q(n-1)$ is accessible. The latter issue is solved in Algorithm 3. Here, in line 4, the information of the last time instant is shifted to the next time instant to construct the coupling constraints $g_{q,1}^p, \dots, g_{q,N-1}^p$. The last prediction step is duplicated by using Assumption 1 to obtain $g_{q,N}^p$ in line 6, which is used to show the recursive feasibility of the system. Furthermore, the uniqueness of the solution of the OCP is ensured by selecting a strictly convex stage cost function $\ell_p$.

---

**Algorithm 3** Resolving asymmetry of communication data for robot $p$

---
1: **Given** robot $p$ and communication data $\mathcal{I}_q(n)$ for $q \in [1 : p - 1]$ and $\mathcal{I}_q(n - 1)$ for $q \in [p + 1 : P]$
2: **for** $q = p + 1$ to $P$ **do**
3:     **for** $k = 1$ to $N - 1$ **do**
4:         **Set** $\mathcal{I}_q(n)(k) := \mathcal{I}_q(n - 1)(k + 1)$
5:     **end for**
6:     **Set** $\mathcal{I}_q(n)(N) := \mathcal{I}_q(n - 1)(N)$
7: **end for**
8: **Set** $i_p(n)$ according to Equation (6)

---

Each robot executes Algorithm 2 to perform the DMPC scheme with given initialisation conditions and targets until the target condition is matched.

## 4. Prediction Horizon Length and Convergence

In this section, we derive sufficient conditions on the prediction horizon length to guarantee a decrease of the costs function and practical stability of the overall system. As the property of initial and recursive feasibility is needed to ensure that the algorithm does not terminate unexpectedly and a feasible solution is found for any time instant $n \geq 0$, we recap shortly the assumptions and theorems which were already presented in Reference [10]:

**Assumption 2** (Feasible initial conditions). *Given a set of robots $[1 : P]$, we have that $z_p^0 \in Z$ and for all $p, q \in [1 : P]$ with $p \neq q$ condition $g_{q,0}^p(z_p^0, q(z_q^0)) \geq 0$ holds.*

Assumption 2 states that the scenario is well defined regarding the state and coupling constraints. Based on this assumption, we can conclude the following:

**Lemma 1** (Initial feasibility). *Consider a set of robots $[1 : P]$ with the model in Equation (1) and the constraints in Equations (2) and (3) satisfying Assumption 1. If Assumption 2 holds, then there exists a solution to the OCP in Equation (16).*

Concluding that there exists a solution to the initial optimal control problem (initial feasibility), recursive feasibility ensures that there exists a solution for any following time instant $n > 0$:

**Theorem 1** (Recursive feasibility). *Suppose a set of robots $[1 : P]$ with the underlying model in Equation (1) and the constraints in Equations (2) and (3) satisfying Assumptions 1 and 2. If Algorithm 2 is applied, then the problem is recursively feasible, i.e., for all $n \in \mathbb{N}_0$ and all $p \in [1 : P]$, there exists a solution to OCP in Equation (16).*

The detailed proofs are presented in Reference [10].

### 4.1. Prediction horizon length with an occupancy grid

In Model Predictive Control (MPC), an appropriate prediction horizon length is crucial for convergence. Here, as the constraints utilising the maximum norm and the intermediate safety margin have to be obeyed (Proposition 2), the indifferentiability of the occupied cells has to be taken into account, which requires a suitable horizon length to enable the optimiser to recognise a decrease of costs by taking a detour. The following assumption is stated to ensure disjunct, feasible initial conditions, and targets of the robots:

**Assumption 3** (Disjunct initial conditions/targets). *Consider a set of robots $[1 : P]$ with the underlying model in Equation (1), the state in Equation (2), and the control constraints in Equation (3). For each robot $p \in [1 : P]$, let $z_p^0, z_p^\star \in Z$, and for all $p, q \in [1 : P]$ with $p \neq q$ and $\hat{z}_j \in \{z_j^0, z_j^*\}$ with $j \in \{p, q\}$, we have*

$$g_{q,0}^{p}(\hat{z}_p, q(\hat{z}_q)) \geq 2\overline{\Psi} + c, \tag{20}$$

*and for* $\hat{z}_p = (x_p, y_p)^{\top}$

$$-\bar{x} + 2\overline{\Psi} + c \leq x_p \leq \bar{x} - (2\overline{\Psi} + c), \quad -\bar{y} + 2\overline{\Psi} + c \leq y_p \leq \bar{y} - (2\overline{\Psi} + c) \tag{21}$$

*shall hold.*

This assumption ensures the existence of a connected feasible path between all initialization points and to all targets, i.e., between any two of these points, there is one grid cell available for a third robot such that all constraints are satisfied. Additionally, to allow that other robots could pass at the bounds of $Z$, this is ensured by Equation (21). For an illustration, see Figure 2. Here, $\Psi$ is included in the figure, so the overall distance for Equation (20) leads to $2\overline{\Psi} + c$ to ensure that a robot can still move between the initial conditions or targets of the other robots.

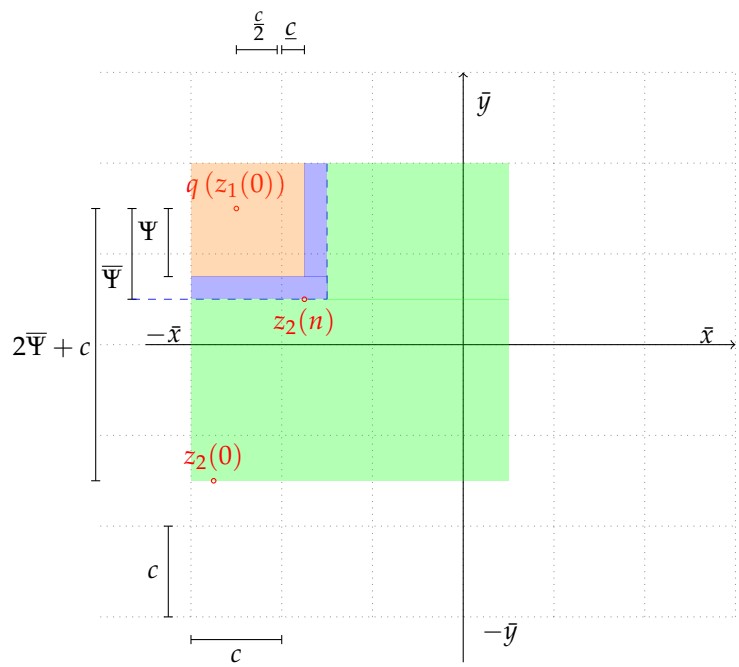

**Figure 2.** Example of a given initial $z_i(0)$, $i \in \{1, 2\}$ and intermediate positions (here, $z_2(n)$) with initial constraints (green) and constraints for intermediate states ($n > 0$, purple and orange) in a grid $\mathcal{G}$ with $a_{\max} = b_{\max} = 3$.

We like to point out that Assumption 3 is stricter as the target positions are included in the feasibility assumption and therefore directly induces Assumption 2 to hold. In order to obtain a sufficient prediction horizon length, which depends on the formulated cost function, a class of stage costs has to be defined:

**Definition 2** (Stage cost function). *: Let* $\ell_p \colon Z \times U \to \mathbb{R}_{\geq 0}$ *be a positive definite, continuous differentiable function with*

$$\ell_p(z_p, u_p) := \left\| d_z(z_p, z_p^*) \right\|_2 + \lambda \left\| d_u(u_p, u_p^*) \right\|_2 \tag{22}$$

*where* $d_z \colon Z \times Z \to \mathbb{R}_{\geq 0}$ *and* $d_u \colon U \times U \to \mathbb{R}_{\geq 0}$ *describe metrics with* $\ell_p(z_p^*, 0) = 0$, $\ell_p(z_p, u_p) > 0$ *for* $z_p \in Z$ *and* $u_p \in U$ *and where* $0 \leq \lambda \leq 1$, $\lambda \in \mathbb{R}_{\geq 0}$ *describes the fraction of the included penalty of the control.*

For stage cost functions based on a normed distance, we have to evaluate the worst case of a necessary feasible detour caused by circumventing another robot's cell in comparison to the direct path to ensure a sufficient prediction horizon length. Therefore, we consider the trajectory points (start and end point of the detour) of a robot at a cell to be circumvented. Then, as most pairs of start and end points of this detour are on the opposite sides of the cell, which has to be circumvented, these pairs can be classified as shown in Figure 3 by either taking the middle of a side length for start and (intermediate) targets or the diagonal length of an asymmetric pair.

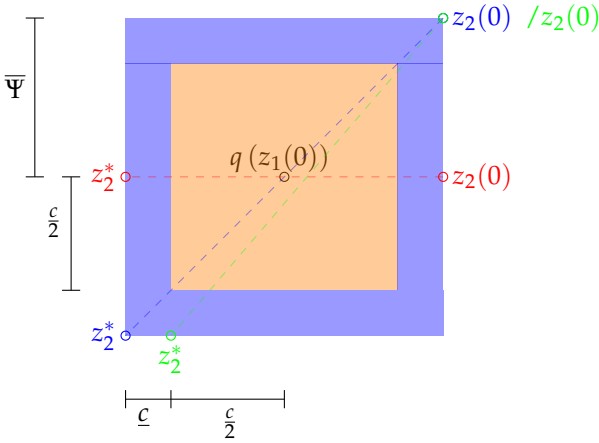

**Figure 3.** Classification of start and end combinations of robot 2 to circumvent robot 1 to obtain the ratios of detour/direct way: middle of side length (red), diagonal (blue), and asymmetric (green).

We state the following theorem to obtain the worst-case positioning of the robots, i.e., we derive a maxima of the ratio of direct path to the necessary detour that a robot has to take:

**Theorem 2** (Worst-case positioning of robots). *Suppose a group of robots with the time discrete model in Equation (1) restricted by the state in Equation (3) and the control constraints in Equation (17) with feasible initial conditions and disjunct (intermediate) targets $z_p(n), z_q(n), z_p(n+N), z_q(n+N) \in Z$ satisfying Assumption 3 with $z_p(n+N) \neq z_q(n+N)$ and a minimum cell size $\underline{c}$ obtained by Equation (9). For two robots $p, q \in [1:P]$, $q$ keeps its position $z_q(k) \equiv z_q$ with $k \in [n:n+N]$. Then, the worst-case of positioning of two robots is bounded by the maximum of the ratio of a necessary feasible detour ($3\left(\frac{c}{2} + \underline{c}\right)$ for moving along the sides of the cell and $\sqrt{\left(\frac{c}{2} + \underline{c}\right)^2 + \delta\left(\frac{c}{2} + \underline{c}\right)^2}$ for the hypotenuse to the target) to the direct path $\|z_p(n) - z_p(n+N)\|_2$ (c.f. $z_2$ (blue) in Figure 4b) by*

$$\frac{3\left(\frac{c}{2}+\underline{c}\right) + \sqrt{\left(\frac{c}{2}+\underline{c}\right)^2 + \delta\left(\frac{c}{2}+\underline{c}\right)^2}}{2\left(\frac{c}{2}+\underline{c}\right) + \delta\left(\frac{c}{2}+\underline{c}\right)} = \frac{3\overline{\Psi} + \sqrt{\overline{\Psi}^2 + \delta\,\overline{\Psi}^2}}{2\overline{\Psi} + \delta\,\overline{\Psi}} = \frac{3\overline{\Psi} + \overline{\Psi}\sqrt{(1+\delta)^2}}{(2+\delta)\,\overline{\Psi}}$$

*with $\delta \to 0$ leading to the maximum ratio*

$$\lim_{\delta \to 0} \frac{3\overline{\Psi} + \overline{\Psi}\sqrt{(1+\delta)^2}}{(2+\delta)\,\overline{\Psi}} = \frac{4\overline{\Psi}}{2\overline{\Psi}} = 2.$$

*This coincides with the shortest feasible target distance (opposite of both sides of the cell of $z_p$, c.f. $z_2$ (red) in Figure 4b) with*

$$\frac{4\left(\frac{c}{2}+\underline{c}\right)}{\|z_p(n) - z_p(n+N)\|_2} = \frac{4\left(\frac{c}{2}+\underline{c}\right)}{2\left(\frac{c}{2}+\underline{c}\right)} \equiv \frac{4\overline{\Psi}}{2\overline{\Psi}} = 2, \tag{23}$$

*and therefore, the ratio of shortest feasible detour to direct path is bounded by*

$$1 \leq \frac{3\overline{\Psi} + \overline{\Psi}\sqrt{(1+\delta)^2}}{(2+\delta)\,\overline{\Psi}} \leq \frac{4\overline{\Psi}}{2\overline{\Psi}} \leq 2.$$

**Proof.** Suppose, a larger ratio of detour to direct path exists. Then, we have to select a combination of start and end points on the side length of the cell. In total, four combinations can be classified, which are depicted in Figure 3. Setting the diagonal opposite edges (Figure 3, blue pair) would lead to a ratio of detour to direct way of

$$\frac{4\left(\frac{c}{2} + \underline{c}\right)}{\|z_p(n) - z_p(n+N)\|_2} = \frac{4\left(\frac{c}{2} + \underline{c}\right)}{\sqrt{4\left(\frac{c}{2} + \underline{c}\right)^2 + 4\left(\frac{c}{2} + \underline{c}\right)^2}} = \frac{4\overline{\Psi}}{\sqrt{8\overline{\Psi}}} = \sqrt{2},$$

and therefore, this is a contradiction to the hypothesis. If a nonsymmetric pair is chosen (Figure 3, green pair) with $0 \leq \delta \leq 1$, this lead to a ratio with

$$\frac{2\left(\frac{c}{2} + \underline{c}\right) + 2\delta\left(\frac{c}{2} + \underline{c}\right)}{\sqrt{4\left(\frac{c}{2} + \underline{c}\right)^2 + 4\delta^2\left(\frac{c}{2} + \underline{c}\right)^2}} = \frac{2\left(\overline{\Psi} + \delta\,\overline{\Psi}\right)}{2\sqrt{\overline{\Psi}^2 + \delta^2\overline{\Psi}^2}} \leq \sqrt{2}, \tag{24}$$

and therefore, all these combinations lead to a ratio of detour to direct tour of less then 2 and, therefore, a larger ratio then in Equation (23) does not exists, which completes the proof. □

Theorem 2 states the worst-case ratio, which has to be covered by a necessary detour in favour to the direct path, which also determines the ratio for the cost function to let the optimiser recognise a decrease of costs by circumventing a robot. This is illustrated in Figure 4a for two robots, where the quantised position of robot 1 is given as an occupied cell.

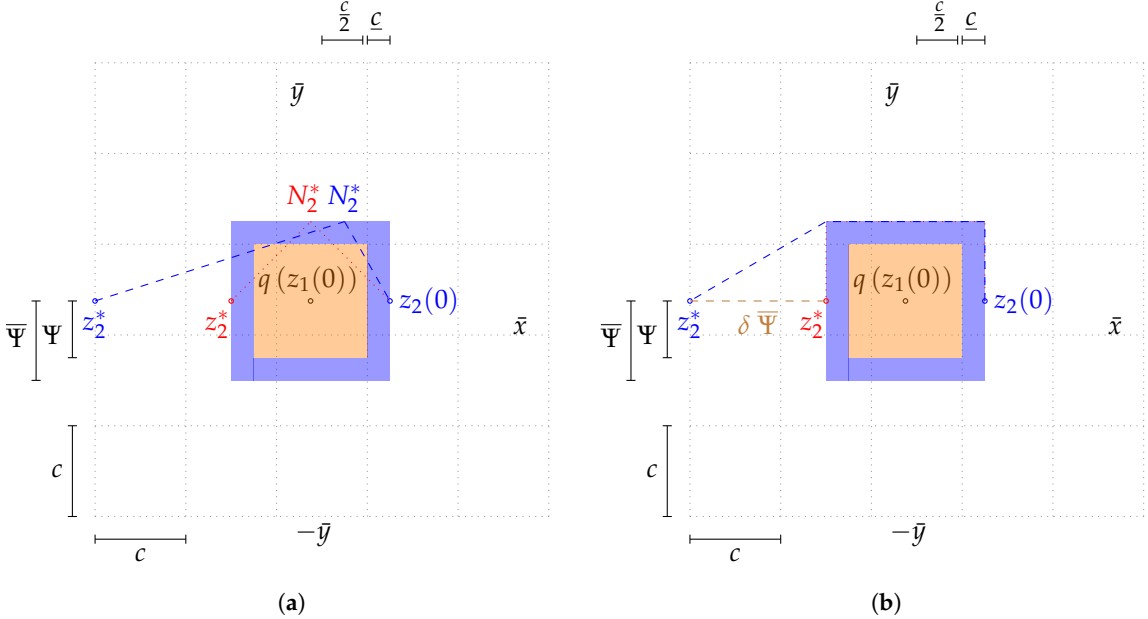

(**a**)　　　　　　　　　　　　　　　　　　　　　　　　　　(**b**)

**Figure 4.** Illustration of the sufficient prediction horizon length $N_2^*$ to recognise a decrease of costs by taking the detour (**a**) and feasible trajectories for robot 2 (**b**), with $x_i(0)$ as the initial position, $q(\cdot)$ as the quantised state, and $x_i^*$ as the (intermediate) target for $i \in [1:2]$.

The sufficient horizon length $N_p^*$, necessary for the optimiser to recognise the decrease of costs, additionally depends on the position of the given target $z_p^*$, illustrated in Figure 4a for robot 2. Examples

for up to 4 robots ($P \leq 4$) are presented in the Appendix. Based on this, we can derive a lower bound for the sufficient horizon length assuming the worst-case positioning from Theorem 2, where the ratio of necessary detour to shortest way is at maximum. An example of feasible trajectories is presented in Figure 4b, illustrating the trajectories depending on the target used for the maximum ratio by Theorem 2.

Note that, although the ratio is bounded, an additional difficulty to determine a lower bound on the prediction horizon length occurs as this depends also on the formulation of the cost function: The costs do not depend solely on the difference of current $z_p(n)$ and target $z_p^*$. A positive definite cost function $\ell_p$ defined by Definition 2 may include a penalty on the control with $0 < \lambda \leq 1$ to avoid large control values or large differences in the control values as many cost functions aiming for a set point are defined for the euclidean space according to Reference [32] [Equation 3.3]. The open-loop costs for a robot retaining its position over the prediction horizon $N$ are given by $(N-1)\,\ell_p\left(z_p(n; z_p^0), 0\right)$. Then, if such a robot holds this position, the costs to circumvent the occupied cell have to be lower, i.e.,

$$(N-1)\,\ell_p\left(z_p(n; z_p^0), 0\right) > \sum_{k=0}^{N-1} \ell_p\left(z_p(n), u_p(n)\right) \tag{25}$$

holds. Here, a prediction horizon length, which is long enough that a decrease of state costs ($\lambda = 0$) is recognised by choosing a detour, is not necessarily long enough if control effort is also penalised ($\lambda > 0$). Hence, a sufficient prediction horizon length has to incorporate the control effort, and therefore, the whole cost function has to be considered. Using Definition 2 and the assumption that a minimiser for the OCP in Equation (16) exists, the sufficient horizon length to recognise a decrease of the open-loop costs is stated in the following theorem:

**Theorem 3** (Sufficient prediction horizon length). *Suppose that two robots $p, q$ are given with feasible initial conditions $z_{p,q}^0 \equiv z_{p,q}(n)$ and disjunct (intermediate) targets $z_p(K) \in Z$ with $K \in \mathbb{N}_0$, $K < \infty$ satisfying Equations (2) and (3), minimising a cost function defined by Definition 2 with Assumptions 1 and 2 to hold and a feasible trajectory between $z_p^0$ and $z_p(K)$ exists bounded by Theorem 2. If*

$$N_p \geq N_p^* > 1 + \frac{\sum_{k=0}^{N_p^*-1} \ell_p\left(z_p(k; z_p^0), u_p^*(k)\right)}{\ell_p\left(z_p\left(0; z_p^0\right), 0\right)} \tag{26}$$

*holds, then robot $p$ is able to obtain a feasible control, which allows to decrease the costs according to Definition 2 and therefore is able to circumvent robot $q$.*

**Proof.** Based on Assumption 2 and Theorem 2, each robot $p$ is equipped with a feasible (intermediate) initial condition and (intermediate) target, i.e., these are feasible and not occupied by other robots for the given time interval $k \in [n : K]$. By Theorem 2 a feasible trajectory exists between $z_p(n)$ and $z_p(n+N)$ for $N \to \infty$ and bounded costs as the maximum ratio of necessary detour and direct path is bounded. As robot $p$ is steered to $z_p(N) = z_p(K)$ with $K < \infty$ for a large horizon $N$, most of the open-loop cost values of the prediction converge to $\ell_p(z_p(N), 0) \to 0$. With the constrained control in Equation (3), the state $z_p(K)$ is reachable by incorporating the impact of the control with

$$N_p^* \ell_p\left(z_p\left(0; z_p^0\right), 0\right) \leq 1 + \sum_{k=0}^{N_p^*-1} \ell_p\left(z_p(k; z_p^0), u_p^*(k)\right) \leq \max_{u \in U} \sum_{k=0}^{N_p^*-1} \ell_p\left(z_p\left(k; z_p^0\right), u\right),$$

while the upper bound is not necessarily optimal in terms of minimal control effort. Hence, as the states of $z_p(n), z_p(K)$ satisfy Theorem 2 and a minimiser for the OCP in Equation (16) exists with a bounded maximal detour, a feasible trajectory exists for a finite horizon $N$. Therefore, the costs are bounded by the constrained control, which defines the required minimum cell size in Equation (9).  □

The sufficient horizon length is illustrated in the following two examples depicting holonomic and non-holonomic robots, whereas we follow the definition that a robot is holonomic if the total degrees of freedom are equal to the controllable degrees of freedom. Here, we apply cost functions based on Reference [10], and for comparison reasons, the maximum norm constraint definition is used for both systems in Equation (13). The considered cellsize is $c = 2.0$, and the sampling size is $T = 1.0$.

**Example 1** (Holonomic example). *Let be two robots defined by*

$$z^+ = f(z,u) = z + u \text{ with } z_p(0) = \begin{pmatrix} 0 \\ 0 \end{pmatrix}, z_p^* = \begin{pmatrix} -4 \\ 0 \end{pmatrix} \text{ and } z_q(0) = \begin{pmatrix} -2 \\ 0 \end{pmatrix}$$

*satisfying the constraints in Equations (2) and (3) and the control bounded by $u \in [-1,1]^2$ with $u = \begin{pmatrix} u_1 \\ u_2 \end{pmatrix}$, $\|u\| \le \sqrt{2}$ and stage costs*

$$\ell_p(z_p, u_p) := \left\| \begin{pmatrix} (x_p - x_p^*)^2 \\ 5(y_p - y_p^*) \end{pmatrix} \right\|_2 + 0.2 \left\| u_p^2 \right\|_2. \tag{27}$$

*The costs of the open-loop prediction are presented in Table 1, illustrating the sufficient prediction horizon length to circumvent robot q with $N = 7$. The two columns at first show the costs for the open-loop prediction including the control impact with $\lambda = 0.2$ and $\lambda = 0$, respectively. The last column represents the stage costs without imposing any control. For an insufficient horizon length, i.e., $N = 6$, the solution of the optimiser reveals $\mathbf{u}_p = 0$. However, to calculate the costs for taking the detour, the optimal control sequence is calculated with the sufficient horizon length $N = 7$ and stripped by the last value.*

**Table 1.** Open-loop costs with and without control impact.

| $N$ | $\sum_{k=0}^{N} \ell_p \left( z_p \left( k; z_p^0 \right), u_p^*(k) \right), \lambda = 0.2$ | $\sum_{k=0}^{N} \ell_p \left( z_p \left( k; z_p^0 \right), u_p^*(k) \right), \lambda = 0$ | $\sum_{k=0}^{N} \ell_p \left( z_p \left( k; z_p^0 \right), 0 \right)$ |
|---|---|---|---|
| 6 | 102.484 | 102.184 | 96.97 |
| 7 | 112.732 | 112.469 | 113.135 |

*Therefore, if the control is included in the cost function, the prediction horizon has to be at least $N \ge 7$ to recognise the decrease of costs in spite of the necessary detour. For an illustration, see Figure 5a.*

**Example 2** (Non-holonomic example). *Based on Reference [33] an example of a non-holonomic robot defined in the 2D plane via*

$$z^+ = f(z,u) = \begin{pmatrix} x \\ y \\ \theta \end{pmatrix} + \begin{pmatrix} \cos(\theta) \\ \sin(\theta) \\ 0 \end{pmatrix} v + \begin{pmatrix} 0 \\ 0 \\ 1 \end{pmatrix} \omega, \quad z_p(0) = \begin{pmatrix} 0 \\ 0 \\ \frac{\pi}{4} \end{pmatrix}, z_p^* = \begin{pmatrix} -4 \\ 0 \\ 0 \end{pmatrix} \text{ and } z_q(0) = \begin{pmatrix} -2 \\ 0 \\ 0 \end{pmatrix}$$

*with $\theta$ representing its orientation. Moreover, we suppose the given stage costs*

$$\ell_p(z_p, u_p) := \left\| \begin{pmatrix} (x_p - x_p^*)^2 \\ 5(y_p - y_p^*) \\ (\theta_p - \theta_p^*)^2 \end{pmatrix} \right\|_2 + 0.2 \left\| u_p^2 \right\|_2.$$

*Due to the constraints of the kinematic model of the robot, the control might be higher than for the holonomic model if the robot has to turn around. The open-loop costs are given in Table 2, showing the costs with included control impact for $\lambda = 0.2$ and $\lambda = 0$. To compute the control effort for taking the detour for the shorter*

*horizon, the optimal control sequence is computed with the longer horizon with the last value stripped as in the holonomic example.*

**Table 2.** Open-loop costs with and without control impact.

| $N$ | $\sum_{k=0}^{N} \ell_p \left( z_p \left( k; z_p^0 \right), u_p^*(k) \right), \lambda = 0.2$ | $\sum_{k=0}^{N} \ell_p \left( z_p \left( k; z_p^0 \right), u_p^*(k) \right), \lambda = 0$ | $\sum_{k=0}^{N} \ell_p \left( z_p \left( k; z_p^0 \right), 0 \right)$ |
|---|---|---|---|
| 11 | 180.071 | 179.815 | 177.894 |
| 12 | 192.389 | 191.725 | 194.066 |

*Here, the lowest sufficient horizon is $N \geq 12$ to allow the optimiser to recognise the lower costs considering the detour with the additional control impact. For an illustration of the chosen trajetory, see Figure 5b. Therefore, both examples illustrate the necessary lower bound of the sufficient prediction horizon length to recognise a detour with the additional impact by penalising the control. The impact of the control values is too small to change the necessary horizon length but might have impact if a higher penalisation is chosen. A deep analysis about the design of the cost function and control effort analysis to stabilise such a non-holonomic robot is given in Reference [33].*

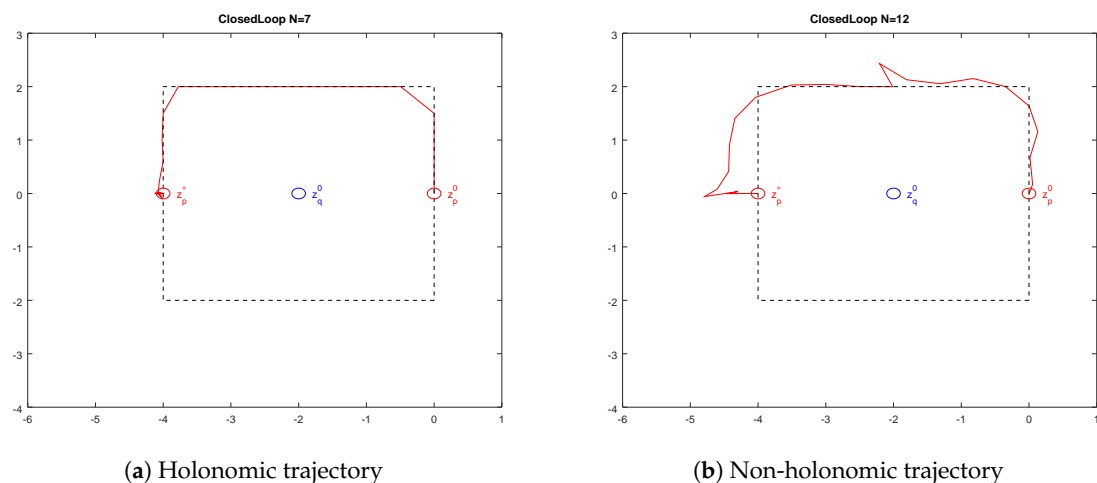

(**a**) Holonomic trajectory        (**b**) Non-holonomic trajectory

**Figure 5.** Holonomic and non-holonomic trajectories of robot $p$ with given initial conditions and targets $z_p(0)$ and $z_p^*$ and of robot $q$ with state $z_q^0$ to be circumvented with the cell constraint illustrated by the dashed rectangle.

## 4.2. Convergence

Utilizing the geometry of our scenario, we show sufficient conditions for the convergence of robots to their given targets. Therefore, we utilise the assumption of feasible initial conditions and targets and the creation of a circle derived from the idea of a roundabout to let the robots use a circular path to ensure deadlock-free execution. An example for the conflicting zone of four robots with corresponding initial conditions and targets is presented in Figure 6.

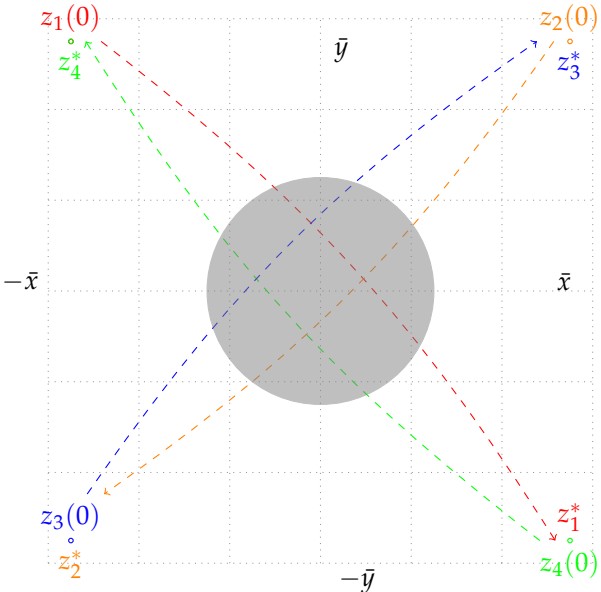

**Figure 6.** Example for a collision avoidance conflict of four robots with given initial conditions $z_i(0)$ and targets $z_i^*$ for $i \in \{1, 2, 3, 4\}$ with $a_{\max} = b_{\max} = 3$ with conflicting area (gray).

**Assumption 4** (Feasible initial conditions and targets allowing boundary values)**.** *Consider a set of robots $[1 : P]$ with the underlying model in Equation* (1)*, the state in Equation* (2)*, and the control constraints in Equation* (3)*. For each robot $p \in [1 : P]$ with $\hat{z}_j \in \{z_j^0, z_j^*\}$, $j \in \{p, q\}$, we have that $z_p^0, z_p^* \in Z$, and for all $p, q \in [1 : P]$ with $p \neq q$, we have that, for all $\hat{z}_p$, a non-empty set $\mathbf{M}(\hat{z}_p) \subset Z$ exists such that*

$$\mathbf{M}(\hat{z}_p) := \left\{ (a, b) \mid (a, b) \in \mathcal{G}, \left\| \hat{z}_p - q\left((a, b)\right) \right\|_2 \leq 2\overline{\Psi} + c \right\}$$

*and*

$$\hat{z}_p \notin \mathbf{M}(\hat{z}_q), \forall q \in [1 : P] \setminus \{p\}, p \neq q.$$

This assumption is weaker than the former one (Assumption 3) to allow for initial conditions and targets close to the boundary of the defined 2D plane that the robots are operating on. For the definition of the roundabout, the creation of the circle is presented in two versions: First, the size of the circle on the total number of robots and, second, a flexible radius is used depending on the number of conflicting robots only. Now, we show under which conditions the convergence of the robots to their targets can be guaranteed.

If the distance of the end of the predictions of two arbitrary chosen robots $i, j \in [1 : P]$ satisfies

$$\| h\left(\mathcal{I}_i(n)(N)\right) - h\left(\mathcal{I}_j(n)(N)\right) \|_2 \leq P\overline{\Psi}, \tag{28}$$

we define

$$(m_x, m_y) = \begin{pmatrix} 0.5\left(m_x^{max} - m_x^{min}\right) \\ 0.5\left(m_y^{max} - m_y^{min}\right) \end{pmatrix} \tag{29}$$

as centre of the circle in planar coordinates and radius $r = \frac{P\overline{\Psi}}{2\pi}$, where

$$
\begin{aligned}
(m_x^{max}, m_y^{max}) = \max \{ & 0.5 \left( h\left(\mathcal{I}_1(n)(N)\right) - h\left(\mathcal{I}_2(n)(N)\right)\right), \ldots, \\
& 0.5 \left( h\left(\mathcal{I}_1(n)(N)\right) - h\left(\mathcal{I}_P(n)(N)\right)\right), \ldots, \\
& 0.5 \left( h\left(\mathcal{I}_{P-1}(n)(N)\right) - h\left(\mathcal{I}_P(n)(N)\right)\right) \} \\
(m_x^{min}, m_y^{min}) = \min \{ & 0.5 \left( h\left(\mathcal{I}_1(n)(N)\right) - h\left(\mathcal{I}_2(n)(N)\right)\right), \ldots, \\
& 0.5 \left( h\left(\mathcal{I}_1(n)(N)\right) - h\left(\mathcal{I}_P(n)(N)\right)\right), \ldots, \\
& 0.5 \left( h\left(\mathcal{I}_{P-1}(n)(N)\right) - h\left(\mathcal{I}_P(n)(N)\right)\right) \}
\end{aligned}
$$

allows to state the equation of the circle. Hence, $(x, y)^\top$ satisfying

$$
(x - m_x)^2 + (y - m_y)^2 = \left( \frac{P\overline{\Psi}}{2\pi} \right) =: r \tag{30}
$$

represents a circle with $x, y \in \mathbb{R}$. To obtain the intersection between the trajectory and the circle, let $\left( x_p(N), y_p(N) \right)^\top = z_p(N)$ and $\left( x_p(N-1), y_p(N-1) \right)^\top = z_p(N-1)$. This allows to state the linear equation with $a^c = y_p(N) - y_p(N-1), b^c = x_p(N) - x_p(N-1), c^c = x_p(N)y_p(N-1) - x_p(N-1)y_p(N)$, which can be inserted into Equation (30), and the intersection between circle and trajectory is obtained by

$$
\begin{pmatrix}
\overline{x}_{1,2} = m_x + \frac{a^c d^c \pm b^c \sqrt{r^2 \left(a^{c2} + b^{c2}\right)}}{a^{c2} + b^{c2}} \\
\overline{y}_{1,2} = m_y + \frac{b^c d^c \mp a^c \sqrt{r^2 \left(a^{c2} + b^{c2}\right)}}{a^{c2} + b^{c2}}.
\end{pmatrix}
$$

To obtain a deadlock-free execution of the collision avoidance mechanism via the circle constraint, the following assumption is required:

**Assumption 5** (Target-point free circle). *For a circle with centre $(m_x, m_y)$ and radius $r = \left( \frac{P\overline{\Psi}}{2\pi} \right)$, suppose that, for all $z_p^*$, $p \in [1:P]$, the conditions $\left| (m_x, m_y)^\top - z_p^* \right| > r$ and $\left| (m_x, m_y)^\top - z_p^0 \right| > r$ hold.*

This assumption ensures that neither an initial condition nor a target $z_p^*$ lie inside the circle. Then, for each robot, the entry point is defined as the intercept point of the circle and the trajectory

$$
z_p^s := \begin{pmatrix} x_p^s \\ y_p^s \end{pmatrix} = \begin{cases} (\overline{x}_1, \overline{y}_1)^\top, & \left\| z_p(N-1) - (\overline{x}_1, \overline{y}_1)^\top \right\|_2 < \left\| z_p(N-1) - (\overline{x}_2, \overline{y}_2)^\top \right\|_2 \\ (\overline{x}_2, \overline{y}_2)^\top, & \text{else} \end{cases} \tag{31}
$$

and the point where the robot leaves the circle is

$$
z_p^e := \begin{pmatrix} x_p^e \\ y_p^e \end{pmatrix} = \begin{cases} (\overline{x}_2, \overline{y}_2)^\top, & \left\| z_p(N-1) - (\overline{x}_2, \overline{y}_2)^\top \right\|_2 < \left\| z_p(N-1) - (\overline{x}_1, \overline{y}_1)^\top \right\|_2 \\ (\overline{x}_1, \overline{y}_1)^\top, & \text{else} \end{cases} \tag{32}
$$

As the radius of the circle is fixed with $r = \frac{P\overline{\Psi}}{2\pi}$, the feasibility of entry points on the circle for each participating robot is guaranteed by Proposition 2. Hence, to keep the robots on the circle, the control law is then changed to

$$
u_p(n) = r \begin{pmatrix} \cos\left( \arccos\left( \frac{x_p(n) - m_x}{r} \right) + u_p^c \right) \\ \sin\left( \arccos\left( \frac{x_p(n) - m_x}{r} \right) + u_p^c \right) \end{pmatrix} \tag{33}
$$

to force the robots on a circular trajectory with $0 \leq u_p^c \leq \underline{c}$ describing the advance of the robot on the circle. As the radius depends on the number of robots, here, $P$, all robots may be driven on a common circle if needed. An illustration of the applied control law is given in Figure 7a.

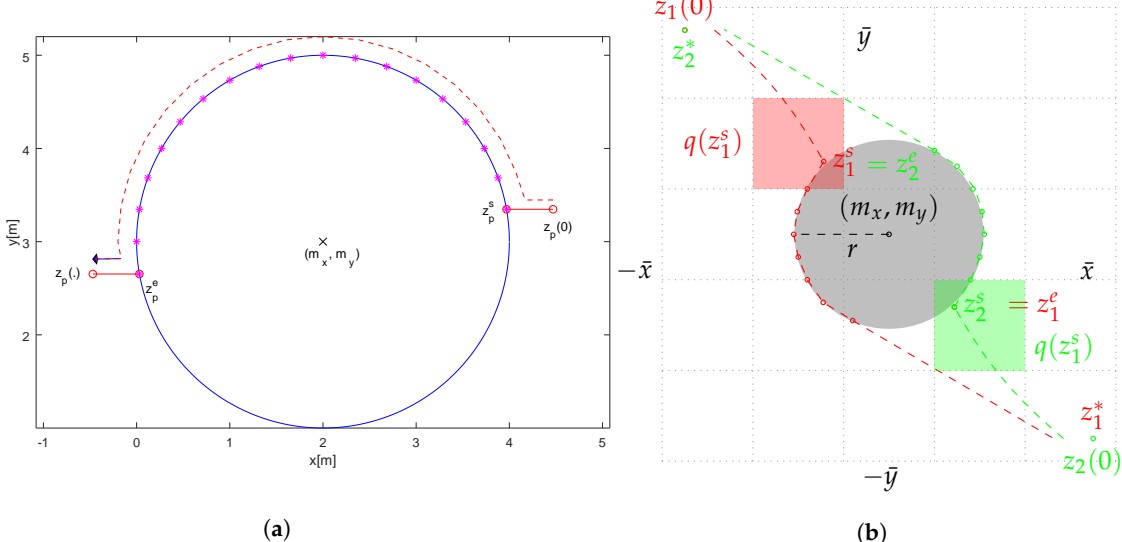

|     |     |
| --- | --- |
| (a) | (b) |

**Figure 7.** Illustration of the circular control with centre $(m_x, m_y)$, and entry and exit points $z_p^s, z_p^e$ for one robot (**a**) and two robots with start and end points $z_1^s, z_2^s$ and $z_1^e, z_2^e$ leaving earlier due to the tangential condition (**b**).

Each robot evaluates the distance to other robots to switch the control law to a circular curve to establish a shared roundabout and to, therefore, ensure the collision avoidance restrictions via Algorithm 4.

If the condition in Equation (28) holds, the algorithm evaluates for each time instant that the predictions of at least two robots undermine their distance. Then, for those robots, the control law is switched and the circular trajectory is carried out for all robots in **Q**, i.e., all robots, which were added to the set **Q**. This set **Q** represents the robots, which satisfy Equation (28) with $2 \leq |\mathbf{Q}| \leq P$. The other robots are set to an immediate hold (Algorithm 4, line 12) and, hence, do not interfere with the created circle. The variable $w_p$ ensures that the evaluation of the centre is carried out once and that the control law is switched back if it was circular before. If the exit $z_p^e$ of the circle is reached or the target $z_p^*$ is tangential reachable (see Algorithm 4, line 18), the remaining trajectory on the circle to $z_p^e$ is skipped and the robot continues under the regime of the original control law, which is set back in Algorithm 4, line 19; see Figure 7b for an illustration. To integrate Algorithm 4 into the DMPC scheme, in Algorithm 2, we include after line 11

**Call** *Algorithm* 4.

Note that, from the circular control law established by Algorithm 4, the robots are still obeying the collision avoidance constraints between the other robots as the DMPC scheme in Algorithm 2 is still executed and Assumption 1 is ensured for $u_p^c$ with $\cos\left(\arccos\left(\frac{z_p(n) - m_x}{r}\right) + u_p^c\right) = 0$ and $\sin\left(\arccos\left(\frac{z_p(n) - m_x}{r}\right) + u_p^c\right) = 0$.

---

**Algorithm 4** Evaluation for switching control law of robot $p$ to establish a collision avoidance circle

---

1: **for all** $p \in [1 : P]$ **do**
2: 　　**if** $\exists q \in [1 : P] \setminus \{p\}$ where Equation (28) holds for $p, q$ **then**
3: 　　　　**Add Q** $:= \mathbf{Q} \cup \{p, q\}$
4: 　　**end if**
5: **end for**
6: **if** $\mathbf{Q} \neq \emptyset$ or $w_p = true$ **then**
7: 　　**if** $w_p = false$ **then**
8: 　　　　**Set** centre $(m_x, m_y)$ for $z_p$ with Equation (29) and $z_p^s, z_p^e \ \forall p \in \mathbf{Q}$ with Equations (31) and
　　　(32)
9: 　　**end if**
10: 　　**Set** $w_p := true$
11: 　　**for all** $p \in [1 : P]$ and $p \notin \mathbf{Q}$ **do**
12: 　　　　**Keep** $u_p = \bar{u}_p \quad \forall k \in [n : n + N]$ according to Assumption 1
13: 　　**end for**
14: 　　**if** $z_p^0 = z_p^s$ and $p \in \mathbf{Q}$ **then**
15: 　　　　**Set** $u_p$ according to Equation (33)
16: 　　**end if**
17: **end if**
18: **if** $w_p = true$ and $\left( \left( z_p^0 - z_p^* \right) \times \left( z_p^0 - (m_x, m_y)^\top \right) = 0 \text{ or } z_p^0 = z_p^e \text{ for } p \in [1 : P] \right)$ **then**
19: 　　**Set** $u_p$ back to Equation (3)
20: 　　**Remove Q** $:= \mathbf{Q} \setminus \{p\}$
21: 　　**Set** $w_p := false$
22: **end if**
23: **if** $\mathbf{Q} = \emptyset$ **then**
24: 　　**for all** $p \in [1 : P]$ **do**
25: 　　　　**Reset** $u_p$ according to Equation (3)
26: 　　　　**Set** $w_p := false$
27: 　　**end for**
28: **end if**

---

**Theorem 4** (Collision avoidance via a fixed circular curve). *Suppose a given number of P robots, each defined by Equation (1), with restrictions on state and control in Equations (2) and (3) and with initial feasible conditions and targets fulfilling Assumption 4. If the distance between two robots is such that Equation (28) is fulfilled and Assumption 5 is satisfied for all robots $p \in [1 : P]$, then the calculation of a circle and execution of the DMPC scheme in Algorithm 2 using the switched control by Algorithm 4 will allow the robots to let them converge to their targets, i.e., to steer the system practically stable.*

**Proof.** From Assumption 4, the feasibility of the initial conditions ($n = 0$) and targets for all $P$ robots is guaranteed, which is weaker than Assumption 3 with additional allowance of points near to the bounds of the 2D plane. Then, with $n > 0$ each robot $p \in [1 : P]$ minimises the distance to the individual target $z_p^*$. Then, if the condition in Equation (28) is fulfilled for at least two robots, the centre $(m_x, m_y)^\top$ of the circle for the conflicting robots is calculated with Equation (29) for the given radius $r$ and the circular control law is applied for all conflicting robots according to Equation (33). All other robots are set to an immediate hold via $\bar{u}_p$. As the condition in Equation (28) holds for any arbitrary chosen robots, the circle is created, when at least two predicted states are closer according to the given condition, and therefore, any other robot is outside this circle fulfilling Assumption 5 and additionally

stops moving. With this assumption that no target is located inside the circle $(z_p^* - (m_x, m_y)^\top)$, each robot $p \in \mathbf{Q}$ is able to calculate a feasible route via Algorithm 4 subject to the intersections $z_p^s$ and $z_p^e$. Furthermore, the circumference of the circle is chosen such that all robots $p \in [1 : P]$ could be integrated on the circular curve and retains the intermediate safety margin, i.e., the trajectories on the circle are feasible. As the exit point is on the opposite position of the circle for each robot, the cost function also decreases with closer distance. Close to the exit point, Algorithm 4 allows an earlier switch to the old control law in Equation (3) if a tangential movement to the original target $z_p^*$ is possible under the condition that orthogonality of the tangent of the current position at the circle and the vector between current position and centre is ensured. Thus, the trajectory still ensures feasibility and allows each robot to converge closer to its target, which renders the system practically stable. □

In a second version, we keep the size of the circle flexible: Instead of using a fixed radius length from Assumption 5, the circle is established for at least two robots with

$$r = (x - m_x)^2 + (y - m_y)^2 = \left( \frac{|\mathbf{Q}| \, \overline{\Psi}}{2\pi} \right) \tag{34}$$

where $\mathbf{Q}$ is the set of robots involved in the circle via Equation (28); therefore $\mathbf{Q}$ reveals $2 \le |\mathbf{Q}| \le P$. Then, the circle is increased in Algorithm 5 as long as no other robot or target is inside the circle with $\left\| z_q(k) - (m_x, m_y)^\top \right\|_2 > r$ for $k \in [n : n + N]$ and $\left\| z_q^* - (m_x, m_y)^\top \right\|_2 > r$, respectively.

Here, the circle is increased in Algorithm 11, line 11 as long as no current position $z_p^0$ or target $z_p^*$ is inside the circle. Again, the variable $w$ ensures that only once the circle is calculated and not changed further. A robot $p$ not participating in this circle switches the control to $u_p = 0$ according to the stop assumption as long as the circle exists. Then, after the robots following the circle switches their control law back according to the leaving condition in Algorithm 5, line 22, i.e, $\mathbf{Q} = \varnothing$, the remaining robots are unblocked, i.e., allowing a control $u_p \ne 0$. According to the previous fixed circle algorithm, the DMPC algorithm is modified by including in Algorithm 2

**Call** *Algorithm* 5

after line 11.

---

**Algorithm 5** Flexible increase of the circular curve

---

1: **for all** $p \in [1 : P]$ **do**

2:      **if** $\exists q \in [1 : P] \setminus \{p\}$ where Equation (28) holds for $p, q$ **then**

3:          **Add Q** $:= \mathbf{Q} \cup \{p, q\}$

4:      **end if**

5: **end for**

6: **if** $\mathbf{Q} \neq \varnothing$ or $w_p = true$ **then**

7:      **Set** $k := |\mathbf{Q}|$

8:      **if** $w_p = false$ **then**

9:          **Set** centre $(m_x, m_y))$ for $z_p \forall p \in \mathbf{Q}$ with Equation (29)

10:          **Set** $w_p := true$

11:          **while** $\forall p : \left\| z_p^* - (m_x, m_y)^\top \right\|_2 > \left( \frac{(k+1)\overline{\Psi}}{2\pi} \right)$ and $\left\| z_p^0 - (m_x, m_y)^\top \right\|_2 > \left( \frac{(k+1)\overline{\Psi}}{2\pi} \right)$ and $k \leq P$ 

     **do**

12:              **Set** $r := \left( \frac{(k+1)\overline{\Psi}}{2\pi} \right)$

13:              **Set** $k := k + 1$

14:          **end while**

15:          **for all** $p \in [1 : P]$ and $p \notin \mathbf{Q}$ **do**

16:              **Keep** $u_p = \overline{u}_p \quad \forall k \in [n : n + N]$ according to Assumption 1

17:          **end for**

18:      **end if**

19:      **if** $z_p^0 = z_p^s, \quad p \in \mathbf{Q}$ **then**

20:          **Set** $u_p$ after Equation (33)

21:      **end if**

22:      **if** $w_p = true$ and $\left( z_p^0 - z_p^* \right) \times \left( z_p^0 - (m_x, m_y)^\top \right) = 0$ OR $z_p^0 = z_p^e \quad \forall p \in \mathbf{Q}$ **then**

23:          **Set** $u_p$ according to Equation (3)

24:          **Set** $w_p := false$

25:          **Set** $\mathbf{Q} = \mathbf{Q} \setminus \{p\}$

26:      **end if**

27:      **if** $\mathbf{Q} = \varnothing$ **then**

28:          **for all** $p \in [1 : P]$ **do**

29:              **Reset** $u_p$ according to Equation (3)

30:          **end for**

31:      **end if**

32: **end if**

---

**Theorem 5** (Collision avoidance via flexible circular curve). *For a given number of P robots with initial conditions fulfilling Assumption 4, if Equation (28) is fulfilled for at least two robots, the calculation of a circle based on Assumption 5 by using Equation (34) for Algorithm 5 lets the robots converge to their individual targets, which ensures practical stability.*

**Proof.** Consider again the Assumption 4 with feasible initial and target conditions, which ensures the intermediate safety margins and allows boundary values. Each robot calculates a feasible trajectory minimising the objective with subject to their target within a finite horizon. Then, if Equation (28) is fulfilled, the circle is established for at least two robots $p, q \in \mathbf{Q}, p, q \in [1 : P]$, of which predictions are

closest among all other robots. As this distance allows to establish a circle at least for these two robots, the increasing radius via Algorithm 5 allows a maximum size of the circle without incorporating the states of the other robots. Then, with switching the control law subject to Equation (33) and the subset of conflicting robots, the deadlock-free execution of this subset $\mathbf{Q}$ is ensured. Then, with the robots following the circular curve under the circular control law, each robot $i \in \mathbf{Q}$ is able to reach the opposite position and is able to leave on the same condition as in Theorem 4 when the target is tangential reachable. Moreover, as the exit point is closer to the target, the costs also decrease. As long as $\mathbf{Q}$ is not empty, the other robots are blocked to prevent them from crossing the circle. Note that, after the circle is dissolved, a minimum distance to other robots which were not participating in the circle is guaranteed by Equation (28) as, for such a robot $q \notin \mathbf{Q}$, the distance is at least $\left\| (m_x, m_y)^\top - z_q \right\|_2 > r$ with $z_q \equiv z_q(k)$ for $k \in [n : n + N]$. Moreover, as the control is set to $u_q = \bar{u}_p$ as long as $\mathbf{Q} \neq \emptyset$, after setting back to the original control law (Algorithm 5, line 29), a feasible solution exists due to Assumption 1. The same procedure to create the circle can be repeated to resolve following conflicts while the robots are approaching to their targets, hence being able to converge to their individual targets, i.e., the system achieves practical stability. □

## 5. Numerical Illustration

Detailed numerical results about the occupancy grid setting were presented in References [10,34] using, for the first, a fixed order of optimising and executing and, for the latter, a priority rule setting. Without enforcing explicitly the rule to reveal a circular curve, we like to point out one scenario with $P = 4$ non-holonomic robots from Reference [34], where a minimum open-loop costs priority rule is used, i.e., the robot with minimum costs over the prediction horizon goes first. We selected here two scenarios using cell sizes $c = 1.5$ and $c = 2.0$ and with start and target positions of the robots opposite to each other in the edges of the operation space. Hence, they have to circumvent each other in the centre. In Figures 8 and 9, the snapshots of the different time instants are illustrated to show the evolution of time of the trajectories and the behaviour of the robots.

In the first scenario depicted in Figure 8 with cell size $c = 2.0$, the large cell size demands a large radius due to the collision avoidance constraints. Robot 0 ("car0") goes at first and, hence, may start with a straight trajectory due to the fixed order and the initialisation phase in Algorithm 1. As all robots start with their initial prediction that they keep their position in the first time instant over the prediction horizon, i.e., ensuring feasibility, robot 0 has the centre of the operation space available to use this at first. The other robots ("car1-3") have to incorporate this trajectory from robot 0 and have to choose detours shown by the coloured predictions, which forms a roundabout in a clockwise manner. The occupied cells, which are reserved by the predictions of the robots, are coloured accordingly to the robots. In later time instants, the trajectories show clearly that, although the order may disadvantage the last robots most, the arrival times are quite close due to the identical behaviour of the sidestepping robots.

With cell size $c = 1.5$ shown in Figure 9, the robots form accordingly a smaller roundabout due to smaller cells and less necessary detours. Similar to the first scenario, robot 0 has the advantage of being enabled to take the centre of the operation space due to the fixed optimisation order. The trajectory especially of robot 3 "car3" shows that the detours are shorter as the robot crosses the centre in a more straight line than in the scenario before. Therefore, without setting implicit conditions on the behaviour of the collision avoidance in both scenarios, the optimiser leads to the pattern of a circular curve, which is also used in street traffic as roundabouts.

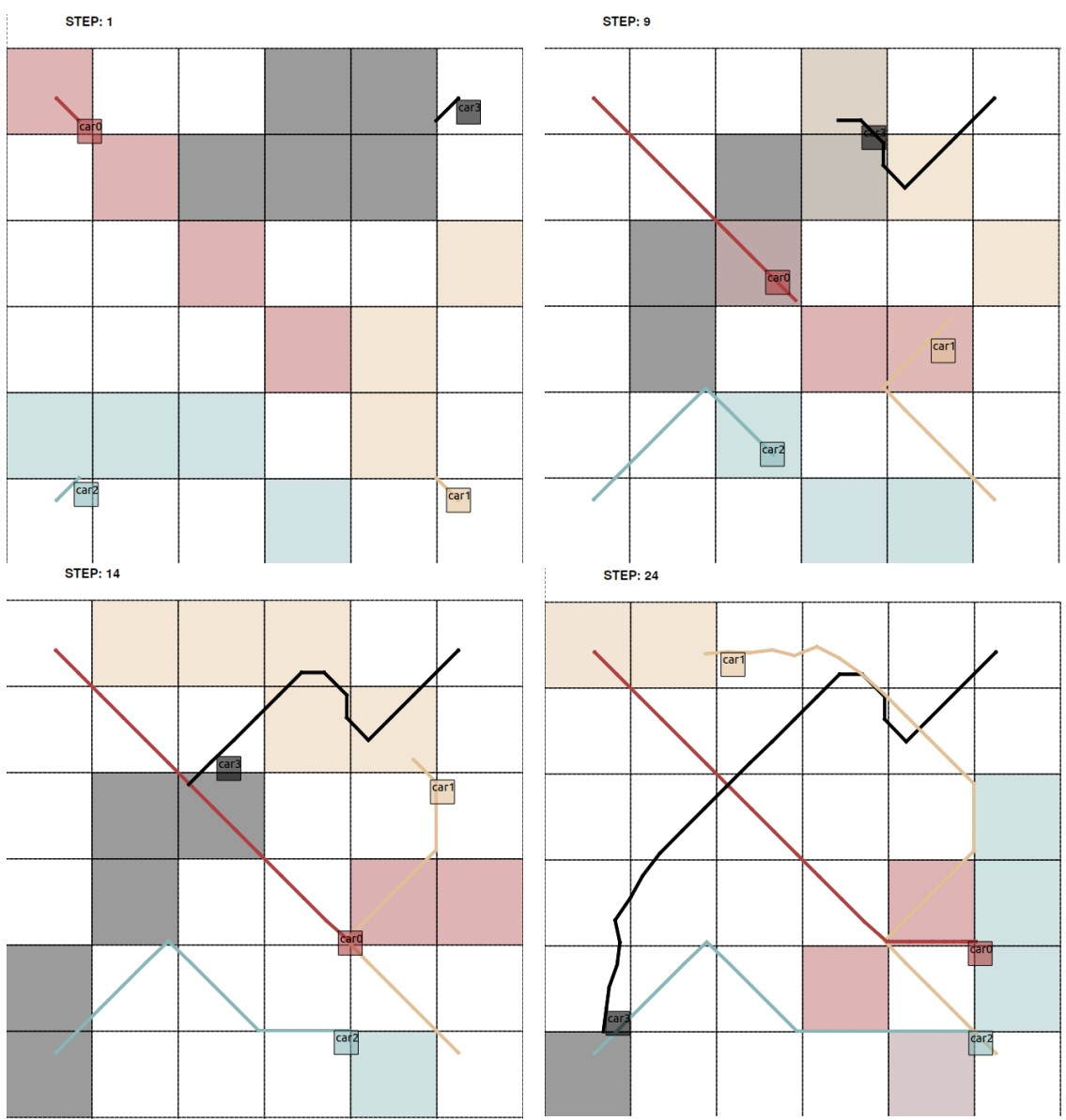

**Figure 8.** Holonomic example with 4 robots, prediction horizon length $N = 12$, and cell size $c = 2.0$: snapshots at time instants $n = 1$ (**left top**), $n = 9$ (**right top**), $n = 14$ (**left bottom**), and $n = 24$ (**right bottom**):Trajectories are drawn in continuous lines, and the predicted occupied cells are coloured in the same colour as the robot.

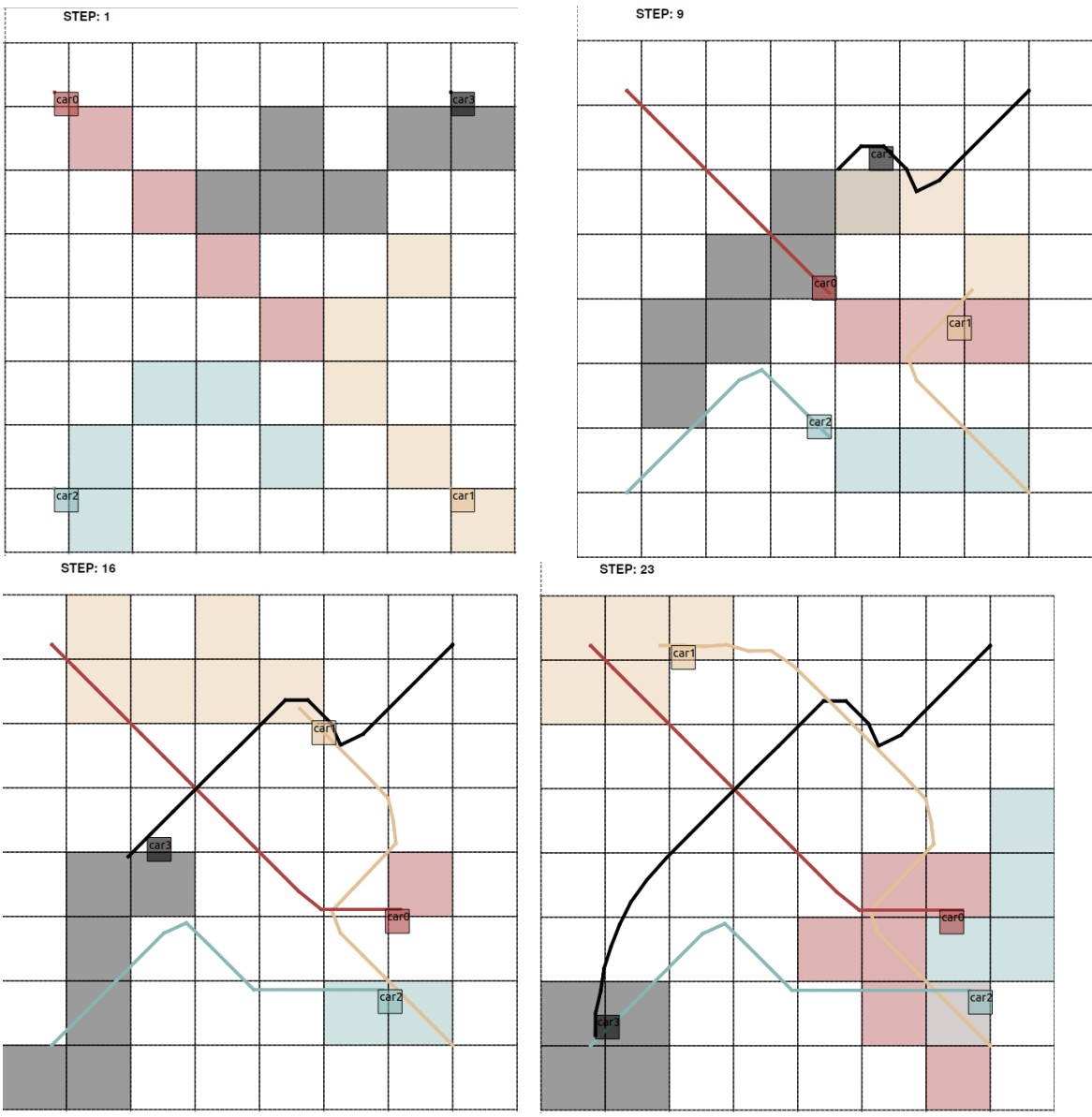

**Figure 9.** Holonomic example with 4 robots, horizon length $N = 12$, and cell size $c = 1.5$: snapshots at time instants $n = 1$ (**left top**), $n = 9$ (**right top**), $n = 16$ (**left bottom**), and $n = 23$ (**right bottom**): Trajectories are drawn in continuous lines, and the predicted occupied cells are coloured in the same colour as the robot.

## 6. Conclusions

In this paper, we derived sufficient conditions for a sufficient prediction horizon length and for convergence to achieve practical stability for a system of distributed robots with quantised communication based on an occupancy grid. We used the properties of the occupancy grid to determine the sufficient prediction horizon length, which is crucial for the optimiser to detect the decrease of costs by taking a detour. The occupancy grid allows to determine the maximum ratio of feasible detour and direct path, which allows to derive such a sufficient prediction horizon length. Moreover, we utilised the idea from a roundabout to establish a control law, which describes a pattern obtained by the numeric results to show convergence for the overall system. This control law was implemented in two versions: First, a fixed radius was used to allow for all robots participating immediately in the circle. Second, this control law was extended to use a flexible radius size to reduce the necessary space to establish the roundabout.

Future considerations include the incorporation of a dynamic optimisation order (priority rules) of the robots and restrictions to traffic scenarios where the freedom of the agents (then cars) is more restricted, e.g., they have to keep within lanes. This may allow easier derivation of conditions to keep such a system deadlock-free, i.e., to steer all distributed agents to their equilibrium.

**Author Contributions:** Conceptualization, T.S. and J.P.; Methodology, T.S.; Software, T.S..; Validation,T.S.; Formal Analysis, T.S. and J.P.; Writing—Original Draft Preparation, T.S.; Writing—Review & Editing, T.S. and J.P.; Visualization, T.S.; Supervision, J.P. All authors have read and agreed to the published version of the manuscript.

**Funding:** This research received no external funding. The APC was funded by the Staats- und Universitätsbibliothek Bremen (SuUB), Germany.

**Conflicts of Interest:** The authors declare no conflict of interest.

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
