# Peer review of "Analytical Aspects of Distributed MPC Based on an Occupancy Grid for Mobile Robots"

_applsci, doi:10.3390/app10031007_

Round 1
Reviewer 1 Report
This paper focused on non-cooperative control of mobile robots based on distributed model predictive control. Following are my comments:
1) What is the difference between cooperative and no-cooperative control. The authors have to mention the difference in the introduction.
2) Why {a_max} and {b_max} are selected to be 4? What are the significance of increasing or decreasing this value.
Reviewer 2 Report
The paper considers a distributed system of non-cooperative
robots with quantized communication based on an occupancy grid,
controlled by a distributed model predictive control scheme.
Some comments and questions:
1) The sentence in the introduction
"In these articles, the convergence or stability aspects were shown
by utilizing a Lyapunov function or a controller, terminal constraints or costs,
a connective constraint or asymptotic controllability."
is not enough clearly. A reformulation is necessary.
2) Although the numerical results are presented in [9,33] it would be useful to include more simulation results.
3) Simulation results shown in Figure 8 are not enough clear and illustrative.
The statement that "the other robots 1-3 circumvents the area in a roundabout pattern" should be illustrated more clearly. Also, it is not clear meaning of colors in Figure 8.
Reviewer 3 Report
The manuscript “Analytical Aspects of Distributed MPC based on an Occupancy Grid for Mobile Robots” is solidly written, sufficiently detailed and seems to be technically correct. I have the following minor comments:
1. This manuscript is quite based on the authors' previous work, especially Sprodowski, T.; Mehrez, M.W.; Worthmann, K.; Mann, G.K.I.; Gosine, R.G.; Sagawa, J.K.; Pannek, J. Differential Communication with Distributed Model Predictive Control of Mobile Robots based on an Occupancy Grid. Information Sciences 2018.
It would be good if the authors further explained and emphasized the contributions of this manuscript compared to previous ones.
2. It seems like the reference [33] has not yet gone through the review process, so it should not be cited as such.
